# A null model of the mouse whole-neocortex micro-connectome

Michael W. Reimann [1], Michael Gevaert[1], Ying Shi[1], Huanxiang Lu [1], Henry Markram[1] & Eilif Muller[1]

In connectomics, the study of the network structure of connected neurons, great advances are being made on two different scales: that of macro- and meso-scale connectomics, studying the connectivity between populations of neurons, and that of micro-scale connectomics, studying connectivity between individual neurons. We combine these two complementary views of connectomics to build a first draft statistical model of the micro-connectome of a whole mouse neocortex based on available data on region-to-region connectivity and individual whole-brain axon reconstructions. This process reveals a targeting principle that allows us to predict the innervation logic of individual axons from meso-scale data. The resulting connectome recreates biological trends of targeting on all scales and predicts that an established principle of scale invariant topological organization of connectivity can be extended down to the level of individual neurons. It can serve as a powerful null model and as a substrate for whole-brain simulations.

[1] Blue Brain Project, Ecole Polytechnique Fédérale de Lausanne (EPFL), CH-1015 Lausanne, Switzerland. Correspondence and requests for materials should be addressed to M.W.R. (email: michael.reimann@epfl.ch)

The study of connectomics has to date largely taken place on two separate levels with disjunct methods and results: macro-connectomics, studying the structure and strength of long-range projections between brain regions, and micro-connectomics, studying the topology of individual neuron-to-neuron connectivity within a region. In macro-connectomics, the absence or presence and strength of projections between brain regions are measured using for example, histological pathway tracing, retrograde[1,2] or anterograde[3] tracers, or MR diffusion tractography[4,5]. While recent advances made it possible to turn such data into connectome models with a resolution of 100 μm[6], this is still far away from single-neuron resolution.

In micro-connectomics, two complementary approaches prevail: stochastic models and direct measures of synaptic connectivity using, for example, electron microscopy. The first uses biological findings to formulate principles that rule out certain classes of wiring diagrams and prescribe probabilities to the remaining ones, while with electron microscopy, snapshots of individual biological wiring diagrams are taken[7–13]. However, published reconstructed volumes at this point only contain incomplete dendritic trees, and therefore incomplete connectivity.

To gain a full understanding of, for example the role of an individual neuron or small groups of neurons in a given behavior, we will have to integrate the advantages of both scales: single-neuron resolution on a whole-brain or at least whole-neocortex level. This has been recognized before[14], but steps toward this goal have until now remained limited. At this point, electron-microscopic reconstructions at that scale are not viable, leaving only statistical approaches to dense micro-connectivity, based on identifying biological principles in the data. Scaling it up to a whole-neocortex level will amplify the uncertainty about the biological accuracy of the results, as many of the resulting connections will be between rarely studied brain regions with little available biological data. Nevertheless, it can serve as a first draft micro-connectome defining a null model to compare and evaluate future findings against. It will also allow us to perform full-neocortex simulations at cellular resolution to gain insights, as to which brain function can or cannot be explained with a given connectome.

We have completed such a first-draft connectome of mouse neocortex by using an improved version of our previously published circuit and connectivity modeling pipeline[15]. It has been improved to place neurons in brain-atlas defined 3d spaces instead of hexagonal prisms, taking into account the geometry and cellular composition of individual brain regions. However, this did not include long-range connections between brain regions, especially the ones formed via projections along the white matter. We therefore set out to identify possible principles, hypotheses of rules constraining the long-range connectivity, and develop stochastic methods to instantiate micro-connectomes fulfilling them.

A first constraint was given by the data on macro- or mesoscale connectivity, which is often reported as a region-to-region connection matrix, yielding a measure proportional to the total number of synapses forming a projection between pairs of brain regions[1,14,16,17]. We used for this purpose, the recently published mesoscale mouse brain connectome of Harris et al.[3]. This data set splits the mouse neocortex into 86 separate regions (43 per hemisphere) and further splits each region when considered as a source of a projection into five individual projection classes, by layer or pathway (Layer 23IT, Layer 4IT, Layer 5IT, Layer 5PT, and Layer 6CT). IT refers to intratelencephalic projections, targeting the ipsilateral and contralateral cortex and striatum; PT refers to pyramidal tract projections, predominantly targeting subcortical structures, but also ipsilateral cortex; CT refers to corticothalamic projections. From here on, we will leave out this additional distinction for projections from layers 2/3, 4, and 6, where only one class is specified in the data of ref. [3]. While the data set does not include GABAergic projection neurons[18], it provides the most comprehensive information on connection strengths of individual projection classes to date.

We further constrained the spatial structure of each projection within the target region. Along the vertical axis (orthogonal to layer boundaries), this was achieved by assigning a layer profile to each projection, as provided by Harris et al.[3]. Along the horizontal axes, we assumed a generalized topographical mapping between regions, parameterized using a voxelized (resolution 100 μm) version of the data provided by Knox et al.[6].

As a final constraint, we applied rules on the number and identity of brain regions innervated by individual neurons in a given source region. To this end, we analyzed the brain regions innervated by individual in vivo reconstructions of whole-brain axons in a published data set (MouseLight project at Janelia, mouselight.janelia.org[19]). Based on the analysis, we conceptualized and parameterized a decision tree of long-range axon targeting that reproduced the targeting rules found in the in vivo data. This approach was generalized to other brain regions for which few or no axonal reconstructions are available.

Finally, we implemented a stochastic algorithm that connected morphologically detailed neurons in a 3d-volume representing the entire mouse neocortex. Synapses were placed onto the dendrites of target neurons according to all the derived constraints by a modified version of a previously used algorithm[15]. Analyzing the results, we found that the constraints we added on top of the region-to-region projection matrices led to a surprisingly complex and non-random micro-structure of neuron-to-neuron connectivity. We characterized this structure to be an extension of an established principle of hierarchical organization of modular connectivity[20] to the level of individual neurons.

## Results

**Neuronal composition and local connectivity**. We placed around 10 million morphological neuron reconstructions in a 3d space representing the entirety of a mouse neocortex. Neuron densities and excitatory to inhibitory ratios at each location were taken from a voxelized brain atlas[21], which is consistent with version 3 of the brain parcellation of the Allen Brain Atlas[22,23]. The composition in terms of morphological neuron types was as in Markram et al.[15]. Reconstructed morphologies were placed in the volume according to densities for individual, morphologically defined subtypes, and correctly oriented with respect to layer boundaries.

For simplicity, we made a strict distinction between local and long-range connectivity, defining local connectivity to comprise any connection where source and target neuron were in the same brain region according to the parcellation in Harris et al.[3], and derived it using previously published methods[24]. All other connections were considered long-range and were derived using the methods described below.

**Constraining the anatomical strengths of projections**. For long-range connectivity, we handled each combination of a projection class (Layer 23, Layer 4, Layer 5IT, Layer 5PT, and Layer 6), a source region, and a target region as a conceptually separate projection. As a first constraint, we determined the average volumetric density of synapses in each projection using published data[3,25], using a programmatic interface provided by the authors. Two further steps were required to apply their data: scaling from projection strength to synapse density, and splitting into densities for individual projection classes.

The biological data provided a measure proportional to the mean volumetric density of projection axons in the target region. Assuming a uniform mean density of synapses on axons across projections, the volumetric synapse density is simply a scaled version of this. We calculated a scaling factor such that the resulting total synapse density in ipsilateral and contralateral projections matches previously published results[26]. From their measured average synapse density ($0.72\,\mu m^{-3}$), we subtracted the synapses we predicted in local connectivity within a region. While a part of the remaining synapses is formed by projections from the hippocampus and extracortical structures, their total number is unclear, but likely comparatively small. For example, the density of synapses in the prominent pathway from VPM into the barrel field[27], when averaged over the whole-cortical depth, is only ~1.5% of the average total density. For now we left no explicit space for synapses from such projections due to the difficulties in parameterizing it for all potential sources.

We then generated matrices of synapse densities for different projection classes by considering projection strengths derived only from tracer experiments in cre-lines associated with a given projection class. Unfortunately, there were no experiments available for some combinations of cre-line and source region. Instead we generated individual matrices by first averaging the reported projection strengths of a line associated with a projection class over modules of several contiguous brain regions (see Supplementary Table 1), and then using that information to generate scaled versions of the wild-type matrix (see the Methods section). Each combination of source and target module were scaled individually, and we enforced the sum of matrices over projection types to be equal to the wild-type. The result is a prediction of the mean volumetric synapse densities from the bottom of layer 6 to the top of layer 1 for all projections (Fig. 1).

**Constraining layer profiles.** So far, we have constrained density and consequently the total number of synapses formed by each individual projection. This reproduces the spatial structure of projections on the macroscale. However, it is likely that there is also spatial structure within a projection, on the mesoscale or microscale. One such structure, acting along the vertical axis is a distinct targeting of specific layers[28]. To constrain the layer profiles of projections, we once more tended to the data published in Harris et al.[3]. The authors provide extensive data on layer profiles, measured hundreds of them, and then clustered them into six prototype profiles using unsupervised hierarchical clustering using spearman correlation and average linkages. As they demonstrate that these prototypes occur in significantly different numbers in feedforward against feedback projections and for the various projection classes and modules, we concluded that they capture sufficient biological detail. We therefore decided to follow this classification and assign one of the prototype profiles to each projection.

Harris et al.[3] already measured the relative frequencies of their prototypical layer profiles for individual projection classes (their Fig. 5o) and for individual source modules, within and across modules (their Fig. 8c, d). They also classified profiles as belonging to feedforward or feedback projections. We combined the constraints by first calculating which layer profiles are overexpressed or underexpressed between pairs of modules, relative to the base profile frequencies for projection classes (see the Methods section). We then classified each projection as feedforward or feedback, based on the hierarchical position of the participating regions, and cut the assumed frequencies of profiles belonging to the other type in half. Finally, we picked for each combination of projection class, source, and target region the layer profile with the highest derived frequency.

We chose to pick the single most likely profile for each projection and ignore the others, as mixing several profiles would have diminished their sharp, distinguishable peaks and troughs. The approach resulted in a prediction, where each profile is used for between 10 and 20% of the projections (Fig. 2a). Based on the prediction, we calculated the resulting relative frequencies of layer profiles per module and per projection class and compared them against the data (Supplementary Fig. 1). We found that in spite of the simplifying step of picking only the most likely profile, the trends in the data were well preserved, although the peaks and troughs were more exaggerated in the model.

We have demonstrated that our simplified predictions recreate the tendencies demonstrated in Harris et al.[3], but the question remains, how do they compare against the raw biological data? As we moved through two consecutive simplifications—from the raw data to six prototypical profiles and from six profiles to a single profile per projection—how much biological detail was lost?

To address this question, we generated raw layer profiles from the voxelized experimental data on projection strength of individual cre-lines[3] using a programmatic interface provided by the authors[6], and compared them to our single prediction. We obtained the connection strength measured with individual tracer experiments in voxels with a resolution of $100\,\mu m^3$ and grouped them by cre-lines associated with the projection classes (see Methods). For each experiment, we calculated a profile of the connection strengths in each layer of a region, relative to the mean across all layers. As a representative example, Supplementary Fig. 2 depicts the model for projections from MOs (blue line) and the data from individual experiments (gray lines). We see an overall fair match between simplified prediction and data, albeit with some errors. For example, the data for 5PT projections show very shallow profiles in four regions that were not predicted. For 5IT projections to visual regions, the data flattens out in layer 1 instead of peaking, although this may be partly artificial, because the data resolution of $100\,\mu m^3$ is close to the width of layer 1, leading to unreliable sampling.

Overall, we find a substantial degree of variability in the biological data, especially for projections from layer 2/3. For example, the density in layer 4 of VISpor due to projections from layer 2/3 of MOs varies between 0.2 and 2.5 times mean. As such, we evaluated the overall match of our predictions relative to the biological variability by calculating the deviation from the biological mean in multiples of the biological standard deviation (z-score, Fig. 2b–f). As a certain number of samples is required to estimate the biological variability, we limited this validation to projections where data from at least five experiments were available. Under the assumption of a Gaussian profile, the data randomly sampled from the biological distribution would follow a standard normal distribution of z-scores (Fig. 2b–f, black dashed lines). We found that the bulk of our predictions fall within that distribution, although a significant number have a z-score exceeding four standard deviations, especially for projections from layer 4 (Fig. 2c). Yet, 75% of z-scores fall within two standard deviations (Fig. 2g).

We conclude that the predicted layer profiles fall within the range of biological variability for most projections, but do result in imperfect densities in individual cases. We judge this to be sufficient for a first draft null model of a white matter microconnectome, but refinement should be attempted in the future, as more data, such as whole-brain axonal reconstructions become available.

**Constraining the mapping of projections.** The previous section constrained projection by imposing a spatial structure along the vertical axis, a layer profile. Yet, it is likely that there is also a

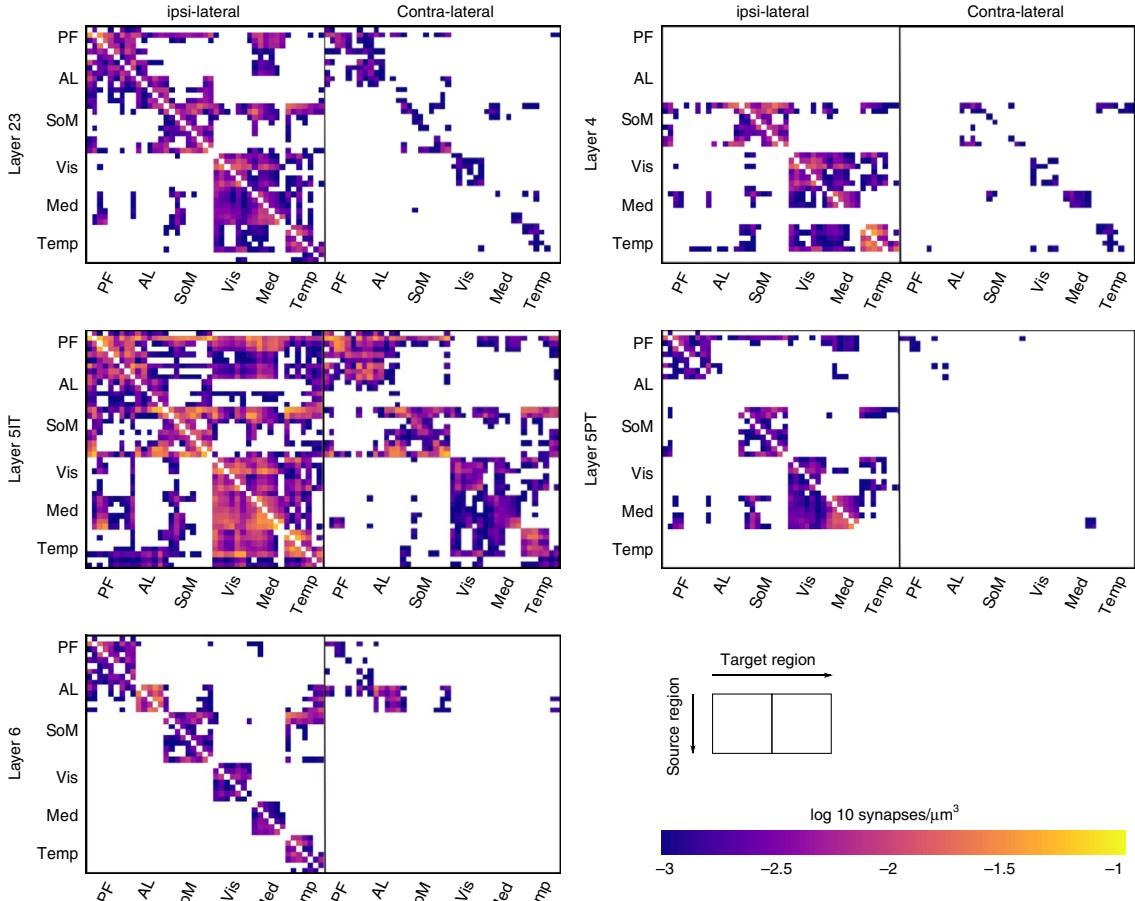

**Fig. 1** Predicted synapse densities in target regions. Modules are labeled: PF: prefrontal, AL: anterolateral, SoM: somatomotor, Vis: visual, Med: medial, Temp: temporal. Exact order of brain regions and assignment to modules by Harris et al.[3] are also listed in Supplementary Table 1. White regions indicate no projections placed for that combination of source and target region

structure along the other two spatial dimensions. That is, that neurons around a given point in the source region project not equally to all points in the target region, but with certain spatial preferences, which we assumed can be expressed by a topographical mapping. To define the mapping, we once more used the voxelized version of the mouse meso-connectome model[6]. As each brain region comprised many voxels in the model, we could use this data to determine whether any given part of a brain region projected more strongly to some part of the target region than to other parts. This would indicate a structured, nonrandom mapping that we would have to recreate to preserve the biologically accurate cortical architecture.

We started by projecting 3d representations of the source and target regions into 2d, preserving distances along the cortical surface (as in Harris et al.[3]). This effectively collapsed the vertical axis, as we had constrained structure along that axis in the previous step. Next, we defined a local barycentric coordinate system in the 2d representation of the source region by picking three points inside the region that maximize the sum of pairwise distances between them, then moving them 25% toward the center. We visualized the result by setting each of the red, green, and blue color channels of an image of the source region to one of the three barycentric coordinates (Fig. 3a, $C_{src}$). By extension, we also associated each voxel of the macro-connectome model ($x$, $y$, $z$) with a color ($B_{x,y,z}$) by first projecting its center into the 2d plane, then looking up the barycentric coordinate. Next, we considered the strengths of projections from each source voxel and visualized the results by coloring each target pixel according

to the product of the 2d-projected projection strength and the color associated with the source voxel:

$$I^{\mathrm{raw}} = f \cdot \sum_{x,y,z \in V_{src}} B_{x,y,z} \cdot F\left(p_{x,y,z}\right), \qquad (1)$$

where $p_{x,y,z}$ refers to the voxelized projection strength from the voxel at $x$, $y$, $z$, $F(p_{x,y,z})$ to its 2d projection and $f$ to a scaling factor effectively deciding the overall lightness of the resulting image. The result is a two-dimensional image with three color channels $\left(I_R^{\mathrm{raw}}, I_G^{\mathrm{raw}}, I_B^{\mathrm{raw}}\right)$. To more clearly reveal the structure of projections, we ignored source voxels associated with a color saturation below 0.5.

The results showed a clear nonrandom structure of targeting in the other regions (e.g., for projections from VISp: Fig. 3b). To parameterize this structure, we first normalized the color values of each pixel, dividing them by the total projection strength reaching that pixel from $src$. We set the denominator to a minimum of 25% of the maximum strength from $src$ in the target region to ensure that weakly innervated parts of $tgt$ would be depicted as such.

$$N_{src}^{tgt}[a,b] = \frac{I^{\mathrm{raw}}[a,b]}{\max\left(I_R^{\mathrm{raw}}[a,b] + I_G^{\mathrm{raw}}[a,b] + I_B^{\mathrm{raw}}[a,b], \sigma_{25}\right)}, \qquad (2)$$

where $I[a,b]$ denotes the pixel of image $I$ at coordinates $a$, $b$ and

$$\sigma_{tgt} = 0.25 \cdot \max_{[a,b] \in tgt}\left(I_R^{\mathrm{raw}}[a,b] + I_G^{\mathrm{raw}}[a,b] + I_B^{\mathrm{raw}}[a,b]\right) \qquad (3)$$

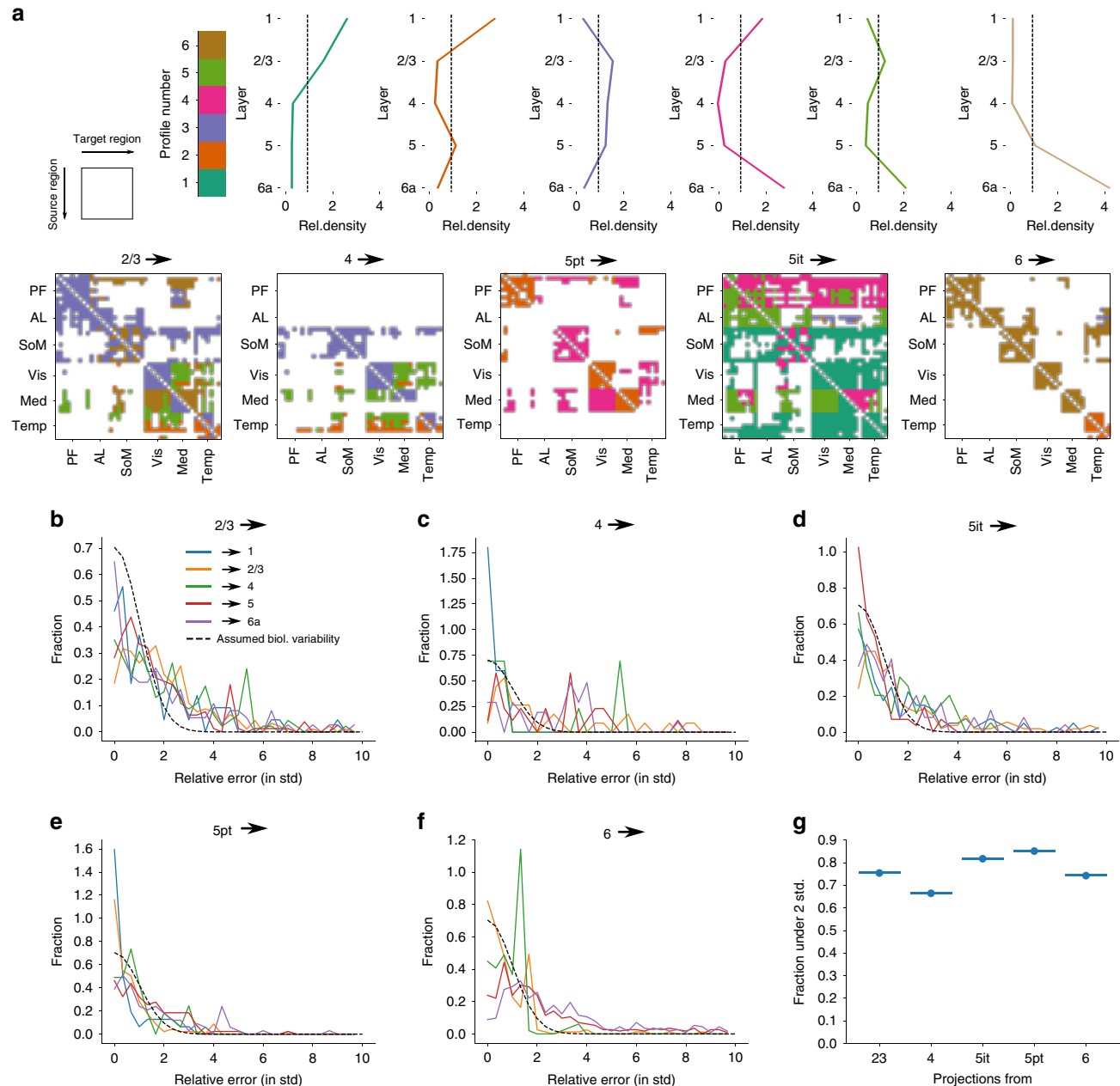

**Fig. 2** Predicted layer profiles. **a** Predictions for all projection classes. Exact order of brain regions and assignment to modules by Harris et al.[3] are also listed in Supplementary Table 1. **b**–**f** Relative error of the predicted synapse densities in all layers. That is, the difference between prediction and the mean of the raw biological data, divided by the standard deviation of the biological data. **b** For projections from L2/3. **c** From L4. **d** From L5IT. **e** From L5PT. **f** From L6. Dashed black lines indicate the biological variability of density under the assumption that it is Gaussian distributed. We used only projections where more than five raw data points to establish the biological variability were available. **g** Fraction of projections where with the relative error under two standard deviations for each source layer

This represented a projection as pixels with normalized lightness, that faded to black in weakly innervated parts of the target region (Fig. 3c, center, $N_{src}^{tgt}$). Next, we optimized a barycentric coordinate system in the 2d-projected target region to most closely recreate the color scheme observed in $N_{src}^{tgt}$ (Fig. 3c, periphery, $M_{src}^{tgt}$). We then assume that a neuron at any coordinate in $C_{src}$ is mapped to neurons at the same coordinate in $M_{src}^{tgt}$. Thus, the two local coordinate systems, each parameterized by three points, together define the topographical mapping between regions src and tgt.

We validated our predicted mapping against established data on the retinotopic mapping in the visual system. This is functional data on the mapping between a brain region and locations in the visual field instead of anatomical data on the projections between brain regions. Yet, we can use it for validation under the assumption that areas corresponding to the same location in the visual field are preferably projecting to each other.

Analyzing the retinotopy, Wang and Burkhalter[29] found certain trends: In adjacent regions, points close to the boundary between them on both sides are mapped together. A counter-clockwise cycle in one area is mapped to a clockwise cycle in an adjacent one. This change in chirality indicates that the mapping must contain a reflection operation. Juavinett et al.[30] utilize this to identify borders between brain areas from intrinsic signal

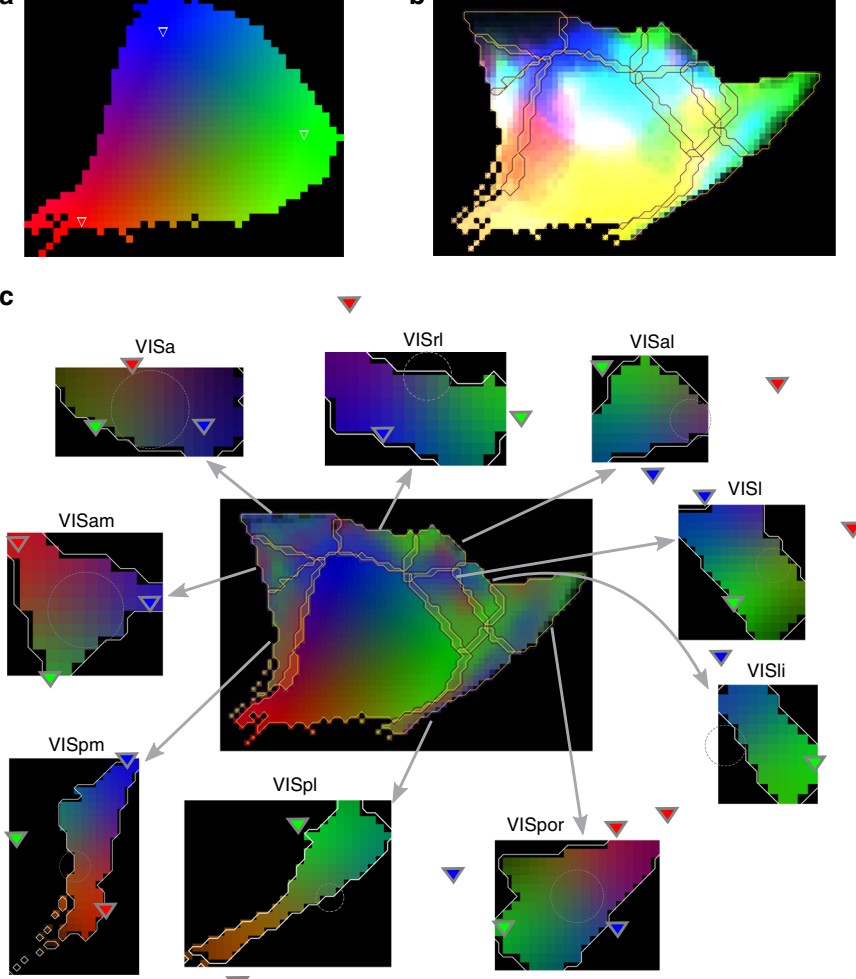

**Fig. 3** Projection mapping in the visual system. **a** The primary visual area (VISp) and its defined source coordinate system. The three points defining the barycentric system are indicated as colored triangles. Each coordinate is associated with the indicated red, green, or blue color channel to decide the color of each pixel in the region. **b** The spatial structure of projections from VISp is indicated by coloring pixels in the surrounding regions according to the color in **a** of the area they are innervated from. **c** Center: as in **b**, but the color of each pixel is normalized such that the sum of the red, green, and blue channels is constant. Periphery: target coordinate systems for the surrounding regions were fit to recreate the color scheme of the center, when colored as in **a**

imaging of retinotopy. When we systematically examined the reflections and rotations in our predicted mapping (Table 1), we found identical results.

Finally, we quantified to what degree barycentric coordinate systems in source and target region can capture the biological trends present in the projection data. As this type of mapping is always continuous and cannot capture nonlinear trends, biological accuracy could be lost. To this end, we calculated the difference between the image of the target region, colored according to the target coordinate system, $M_{src}^{tgt}$, and the normalized image of the target region according to the projection data, $N_{src}^{tgt}$. We defined the relative error of a target coordinate system as the sum of absolute differences of the two images, divided by their average and the number of pixels (Fig. 4). We found that for over half of the projections the error was below 5% and the maximum error was 17%.

**Constraining projection types**. Thus far, we have considered constraints on the spatial structure of projections on a global scale (the macro-connectome matrix) and a local scale (the layer profiles and the mapping). The topographical mapping also limited which individual neurons in a target region can be reached by a given neuron in a source region, severely

constraining the topology of the potential connectome graphs on a local scale. Yet, an important aspect of neocortical connectivity not yet considered is which combinations of regions are innervated by single-source neurons[31]. Even if we know which regions are innervated by a population of neurons in a given region, each individual neuron is likely to innervate only a subset of those regions. We call that subset its projection type or *p-type*. It is unclear to what degree the process is pre-determined or stochastic, and if it is stochastic, what mechanisms further shape and constrain the randomness. This is a complex problem, as a region such as SSp-tr innervates 27 other regions, yielding $2^{27} = 134217728$ potential p-types.

To tackle this problem, we analyzed that reconstructed axons made available by the MouseLight project at Janelia[19]. These are whole-brain neuron reconstructions of cortical neurons that include their long-range projections. We first classified their neuron types, then placed the axons in the context of the Allen Brain Atlas and finally evaluated the amount of axonal length projecting into the 43 ipsilateral and 43 contralateral brain regions.

Figure 5a shows an example of 61 analyzed axons originating in MOs. The scale of the p-type problem is clear at first glance:

only a single combination of innervated regions is repeated in this data set, all others represent unique p-types. Yet, a structure is also apparent: while only 11 out of the 61 axons innervate the

visual or medial modules, the ones that do tend to innervate more than a single of their regions. Moreover, it appears that the projection strength (Fig. 5a, first row) is a strong predictor of the probability that any given axon innervates a region (innervation probability), indicating that a projection is strong because many neurons participate in it, not because of few participating neurons with large axonal trees in the target region.

Next, we analyzed these observations systematically. We only had for the source region MOs a sufficient number of reconstructed axons to robustly estimate the innervation probabilities. We found that innervation probability was proportional to the normalized projection strength, i.e., the amount of axon in the target region, normalized by the volume of the source region. We determined projection class-specific constants of proportionality with a linear fit, resulting in a predicted innervation probability $P = 0.5 \cdot \sqrt{nps}$ for projections from L2/3 and L6, $0.33 \cdot \sqrt{nps}$ from L4 and L5PT and $0.22 \cdot \sqrt{nps}$ from L5IT. Figure 5b compares the innervation probability predicted this way to the one observed in 25 samples for L2/3 of MOs, 61 samples for L5 of MOs, and 35 samples of its L6 ($p = 3 \cdot 10^{-9}$, two-tailed pearsonr, $n = 3 \cdot 86$, i.e., one sample per region x hemisphere x projection class). Conversely, the projection strength was less a predictor of the axon length in a target region for individual axons innervating the region (Fig. 5c). Projection strength being a predictor of innervation probability is in line with the findings of Han et al.[31]. Assuming the principle holds for other brain regions as well, we were able to predict the first-order innervation probabilities for all combinations of source and target region.

### Table 1 Validation of predicted mapping

|  |  | Wang and Burkhalter[29] | Juavinett et al.[30] | Our mapping |
|---|---|---|---|---|
| **VISpl** | reflection? | Yes | Yes | Yes |
|  | rotation? | 90° | n/a | 90° |
| **VISpor** | reflection? | None | None | None |
|  | rotation? | 180° | n/a | 180° |
| **VISl** | reflection? | Yes | Yes | Yes |
|  | rotation? | None | n/a | None |
| **VISli** | reflection? | None | None | None |
|  | rotation? | None | n/a | None |
| **VISal** | reflection? | None | None | None |
|  | rotation? | 180° | n/a | 180° |
| **VISrl** | reflection? | Yes | Yes | Yes |
|  | rotation | None | n/a | None |
| **VISa** | reflection? | None | n/a | None |
|  | rotation? | 90° | n/a | 90° |
| **VISam** | reflection? | None | None | None |
|  | rotation? | 90° | n/a | 90° |
| **VISpm** | reflection? | Yes | Yes | Yes |
|  | rotation? | None | n/a | None |

Comparing linear transformations from source to target coordinate system in our results to the ones of Wang and Burkhalter[29] and Juavinett et al.[30]. n/a indicates that a paper provides no data on a transformation

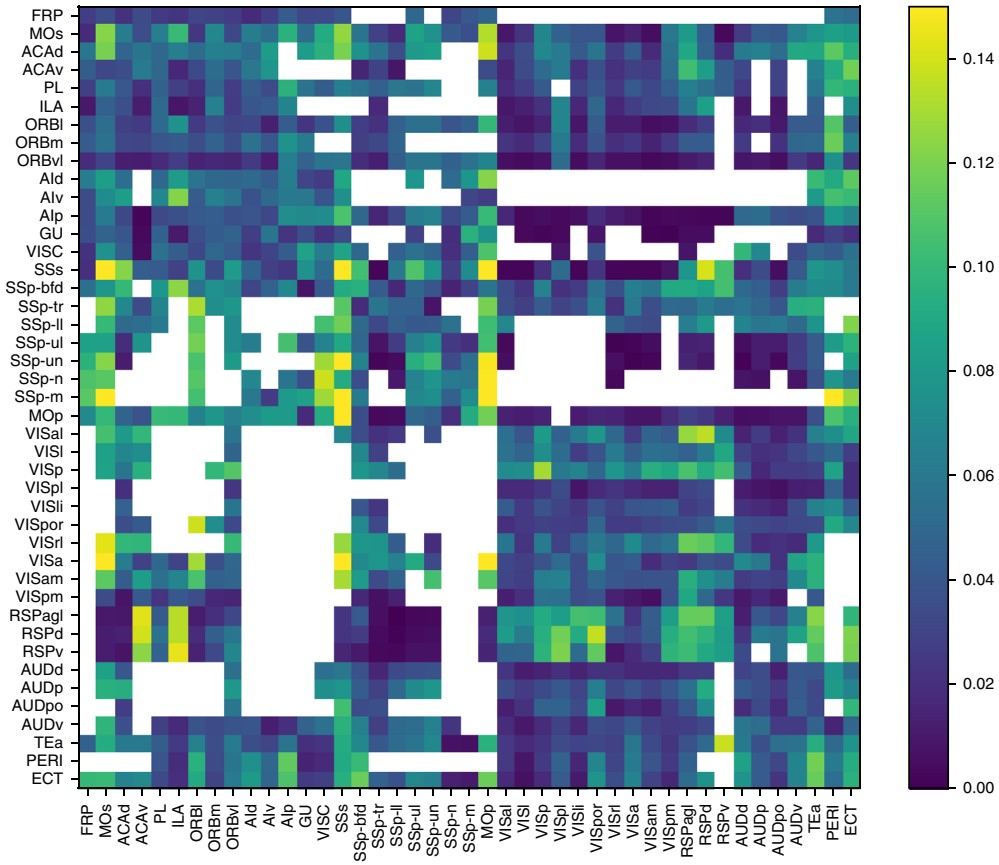

**Fig. 4** Validation of predicted mapping. Relative error of the mapping defined by the barycentric coordinate systems in the target area, compared with the data. Values along the main diagonal: for contralateral mapping; all others: ipsilateral mapping. The data shown where the sum of densities from all projection classes is above 0.025 µm⁻³

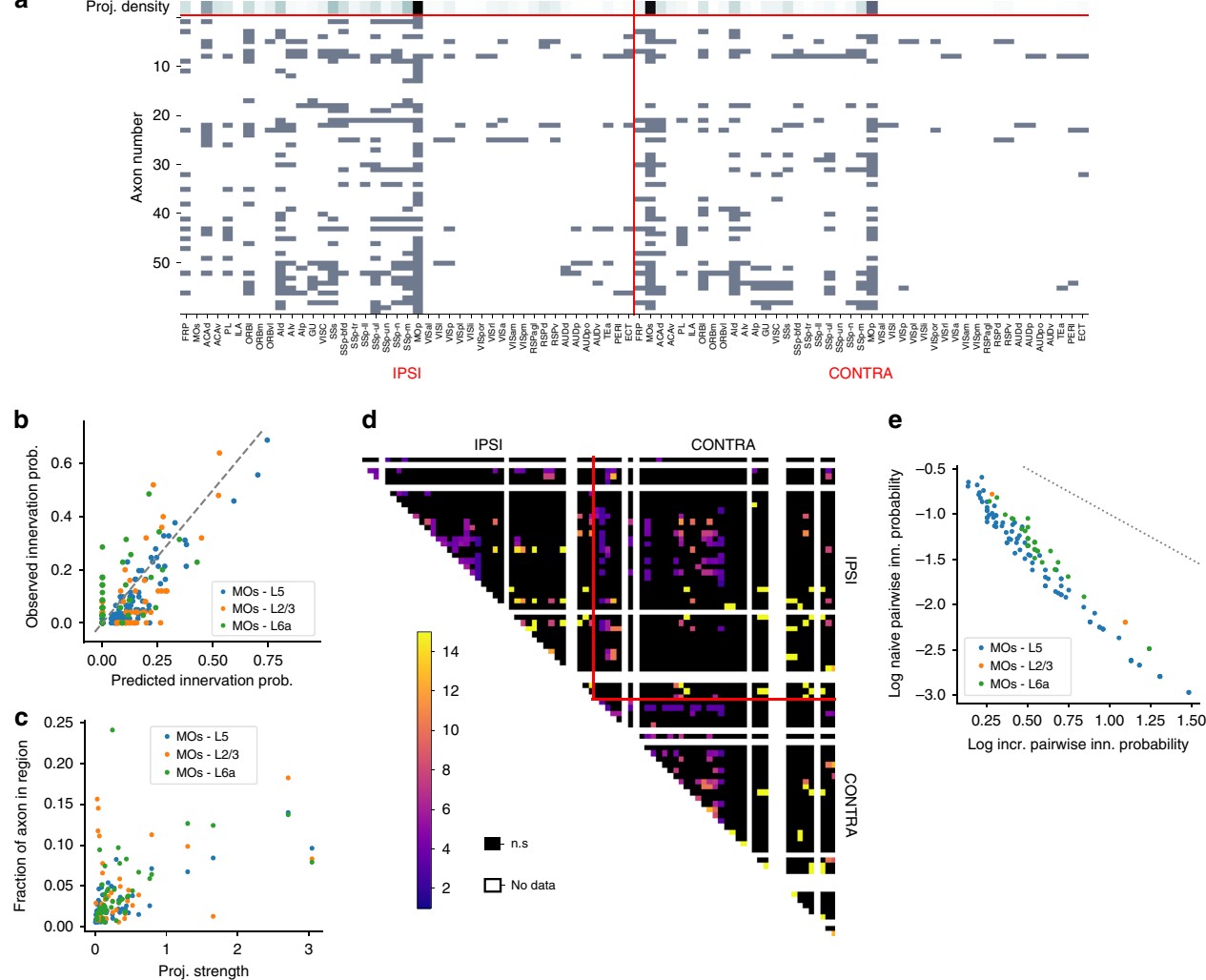

**Fig. 5** Innervation of brain regions by individual axons. **a** Projection density according to Harris et al.[3] (top row), ranging from no projection (white) to strong projections (black), and brain regions innervated by 61 reconstructed axons (rows) indicated by gray squares. **b** Probability to innervate individual brain regions, predicted from the normalized projection strength from MOs, against the observed innervation probability (L2/3: calculated from $n = 25$ axons, L5: $n = 61$ axons, L6a: $n = 35$ axons). **c** Normalized projection strength against the mean total length of axon branches in individual brain regions ($n$ as in **b**). **d** Observed interactions between the innervation of individual brain regions, i.e., increase in innervation probability of one region when the other is known to be innervated. **e** Increase in innervation probability as in **d** against the innervation probability of a pair of regions under the assumption of independence. Gray dotted line indicates the point where the product of independent probability and increase is one that can logically not be exceeded. All innervations and projection strengths in this figure are for projections from MOs

Next, we analyzed statistical interactions of the innervation probabilities for axons originating in MOs. For pairs of target regions, we evaluated the null hypothesis that their innervations are statistically independent, and if it was rejected ($p \geq 0.05$; see the Methods section) calculated the strength of the statistical interaction as the conditional increase in innervation probability $\left( \frac{P(s \to t_1 | s \to t_2)}{P(s \to t_1)} \right)$. We found significant interactions for 283 pairs (Fig. 5d), with some strengths exceeding a 15-fold increase. However, there were several problems preventing us from simply using these observed interactions to constrain connectivity. First, we only had data for axons originating from one of 43 brain regions and it is likely that interactions differ for source regions. Second, the data were incomplete, as some targeted regions were not innervated by a single reconstructed axon (Fig. 5d, white patches), and others were based on only a single or two axons. Third, evaluating $86 \cdot (86-1)/2 = 3655$ potential interactions based on only 61 data points (i.e., axons) are statistically inherently unstable and likely to dramatically overfit.

**A model to generate projection types.** Instead, we tried to use the available axon data to develop a conceptual model of how the interactions arise. We first observed that the largest interactions strengths occurred for target regions in the medial and visual modules that are otherwise only weakly innervated. Evaluating this observation systematically, we found that indeed the strength of an interaction was strongly negatively correlated with the product of the first-order innervation probabilities of the pair (Fig. 5e). Second, we observed only conditional increases in innervation probability (values ≥1), i.e., innervation of pairs of brain regions is not mutually exclusive.

One model explaining both our observations is the following: consider a tree with the brain regions in both hemispheres as the leaves. Let each edge in the tree be associated with a probability that the edge is successfully crossed by an axon, these probabilities can be different in both directions of the edge. To generate the set of innervated regions for a random axon, start at the leaf representing its source region and then consecutively

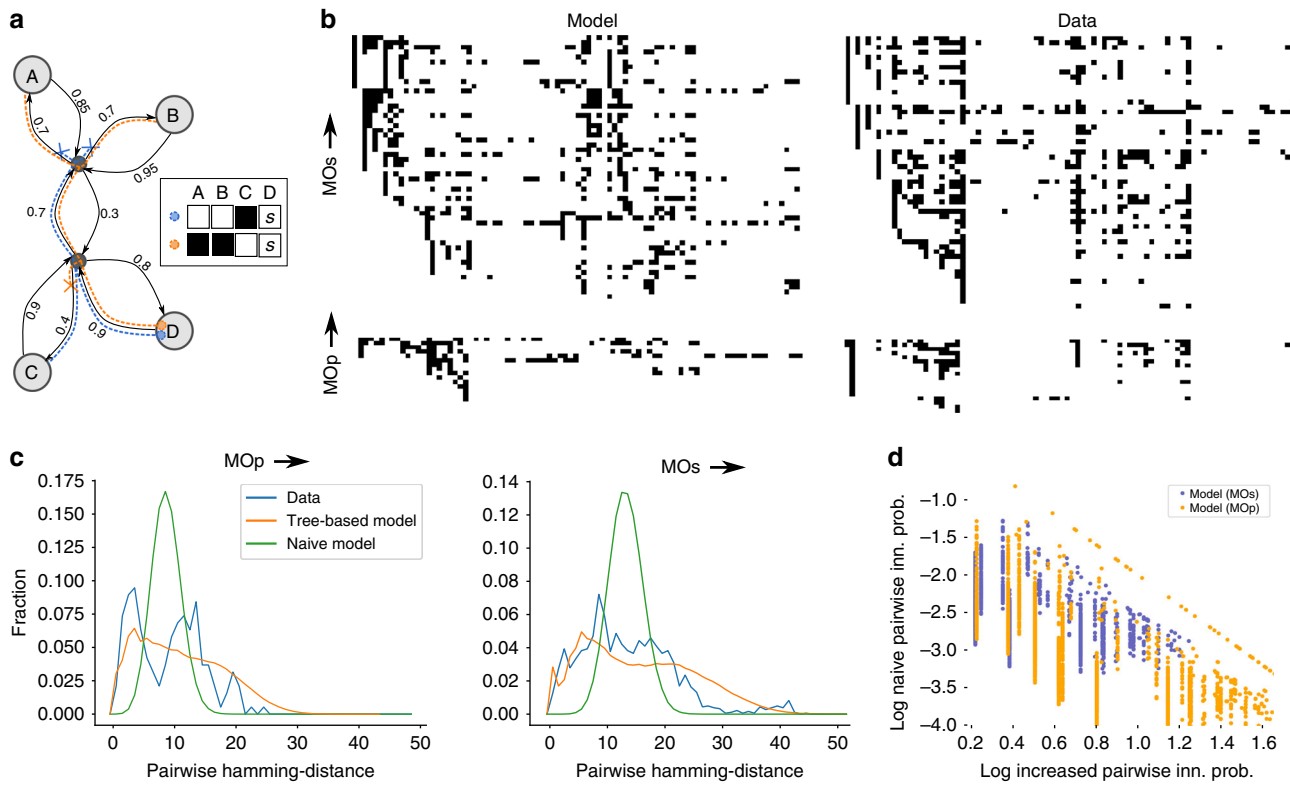

**Fig. 6** A model to generate p-types. **a** Toy example of a p-type generating model with four regions (A–D). The regions are associated with the leaves of a directed tree (black), edges of the tree are associated with a probability to cross it. Two exemplary axons (orange, blue) spread from region D either crossing an edge (dashed lines) or not (dashed X-marks). Inset: resulting p-types; black regions are innervated; s indicates the source region. **b** Examples of innervation of brain regions predicted by the full model for L5IT (left column) and of reconstructed axons (right column). Sampled axons along the y-axis, brain regions along the x-axis. A black pixel indicates that an axon is innervating a region. Top row: axons originating from L5 of MOs; bottom row: from MOp. **c** Pairwise distances (hamming distance) between the profiles of brain region innervation. Blue: the data from reconstructed axons (see **a**); orange: from 10,000 profiles sampled from the tree-based model; green: from 1000 profiles sampled from a naive model taking only the first-order innervation probabilities into account. Left: for axons originating from MOp; right: from MOs. **d** Increase in innervation probability against the basic innervation probability as in Fig. 5e

spread to other nodes further into the tree along its edges with the probabilities associated with the edges (Fig. 6a). Once it has been decided that an edge is not crossed, it cannot be crossed in future steps. Every leaf reached this way is then considered to be innervated by the axon.

If we set the length of an edge in this model to the negative logarithm of the associated probability, then the first-order probability that a region $T$ is innervated by an axon originating in region $S$ is easily calculated:

$$P(S \to T) = 10^{-L(S,T)}, \qquad (4)$$

Where $L(S, T)$ denotes the length of the shortest path between $S$ and $T$.

Similarly, the increase in conditional innervation probability of $T_1$ and $T_2$ is given as:

$$I(S, T) := \frac{P(S \to T_1 | S \to T_2)}{P(S \to T)} = \frac{10^{-L(lca(T_1,T_2),T_2)}}{10^{-L(S,T_2)}}, \qquad (5)$$

Where $lca(T_1, T2)$ is the lowest common ancestor of $T_1$ and $T2$.

Due to the underlying tree structure, the lowest common ancestor is always an inner node that is closer or of equal distance to $T_2$, therefore the strengths of interactions are always larger than one indicating an increase of innervation probability, which is in line with our earlier observations.

Fitting the model consisted of two steps: first, we generated the topology of the tree using the normalized connection density of projections, i.e., the amount of signal (axon) in the target region

normalized by the volume of both source and target region. Specifically, we used the Louvain heuristics[32] with successively decreasing values for the *gamma* parameter to detect successively larger communities in the matrix of normalized connection densities (see Methods). Next, we replaced each edge with two directed edges, one in each direction. Then we optimized the probabilities associated with edges using the first-order innervation probabilities predicted from the normalized connection strength of projections as in Fig. 5b. These predictions then served as constraints on the path lengths between leaves. Specifically, we locally optimized the edges in small motifs consisting of two sibling nodes and their parent, based on differences in the distances of the siblings to all leaves (see Methods). As the pair of edges between nodes can have different associated probabilities, the predicted statistical interactions are not symmetric (see Supplementary Fig. 4) and there can be region-specific differences in the number of regions innervated or innervated from.

We used the fitted model to generate 10,000 profiles of brain region innervation for axons originating from L5 of MOs and MOp. Figure 6b compares a number of randomly picked profiles against the data from reconstructed axons for both regions. As the model was constrained with the predicted first-order innervation probabilities, it manages to recreate the observed high-level trends: strong innervation of the ipsilateral and contralateral prefrontal, anterolateral, and somatomotor modules; weaker, but highly correlated innervation of the other modules.

To test the model further, we calculated the pairwise hamming distances between innervation profiles from reconstructed axons and from the model (Fig. 6c). We also compared the data against a naive model using only the observed first-order innervation probabilities and assuming no interactions. We found that the naive model resulted in a narrow, symmetrical distribution with a single peak at around 9 (MOp) or 13 (MOs). In contrast, the axon data led to a much wider, asymmetrical, and long-tailed distribution that were much better approximated by the tree-based model. The difference between the distribution resulting from the tree-based model and the axon data was, in fact, not statistically significant (MOp: $p = 0.44$, $n = 9$ axons; MOs: $p = 0.12$, $n = 61$ axons; kstest).

Using the tree-model, we could predict the strengths of interactions as described in Eq. (5) (Supplementary Fig. 4). When comparing the strength of the interactions against the naive innervation probabilities without interactions, we found in the model the strong negative correlation that was present in the axon data (Figs. 5e, 6d). For the model, we found more data points toward the lower left corner of the plot that indicates low naive probability and low increase. The lack of such points in the data from axon reconstructions can be explained by the fact that points associated with extremely low probabilities are unlikely to show up in a relatively small sample of reconstructed axons.

As a final validation, we compared the model against the results of Han et al.[31], which considered brain region targeting of single axons originating from VISp. We have not taken into account axons from this source region when we formulated or fitted the model, making this a powerful validation of the generalization power of the model (Fig. 7). Comparing the

number of visual regions innervated (out of VISli, VISl, VISal, VISpm, VISam, and VISrl) by individual axons originating in layer 2/3 of VISp, we find comparable results (Fig. 7a). Although in the model, the mean number of regions innervated is slightly higher (1.84 vs 1.7 (fluorescence-based) or 1.56 (MAPseq)) we find the same roughly binomial distribution where fractions decrease with increasing number of innervated regions.

We were also able to predict this distribution for axons from other layers using our model. We predict similar shapes of the distribution with an even higher mean for layer 5 and a lower mean for layer 4 and especially layer 6. Next, we also considered the statistical interactions between the six visual target regions (Fig. 7b). Again, we found overall comparable conditional probabilities, with a comparable structure, although strong common innervations of regions VISl and VISal and VISpm and VISam were underestimated.

**Connectome instantiations and their micro-structure.** Finally, we developed a stochastic algorithm to generate instances of a neuron-to-neuron connectome that fulfills all constraints in the long-range projection recipe and used it to connect a model of the entire mouse neocortex (see the Methods section). We considered slender tufted and untufted pyramidal cells in layer 5 to participate in projection class L5IT and half of the thick tufted layer 5 pyramidal cells in L5PT, with the other half participating in L5CT, which is not covered by the present, purely cortical model. Pyramidal cells in other layer all participated in the corresponding projection class.

As a result, we obtained connectome instances with 88 billion modeled synapses, each associated with a presynaptic neuron,

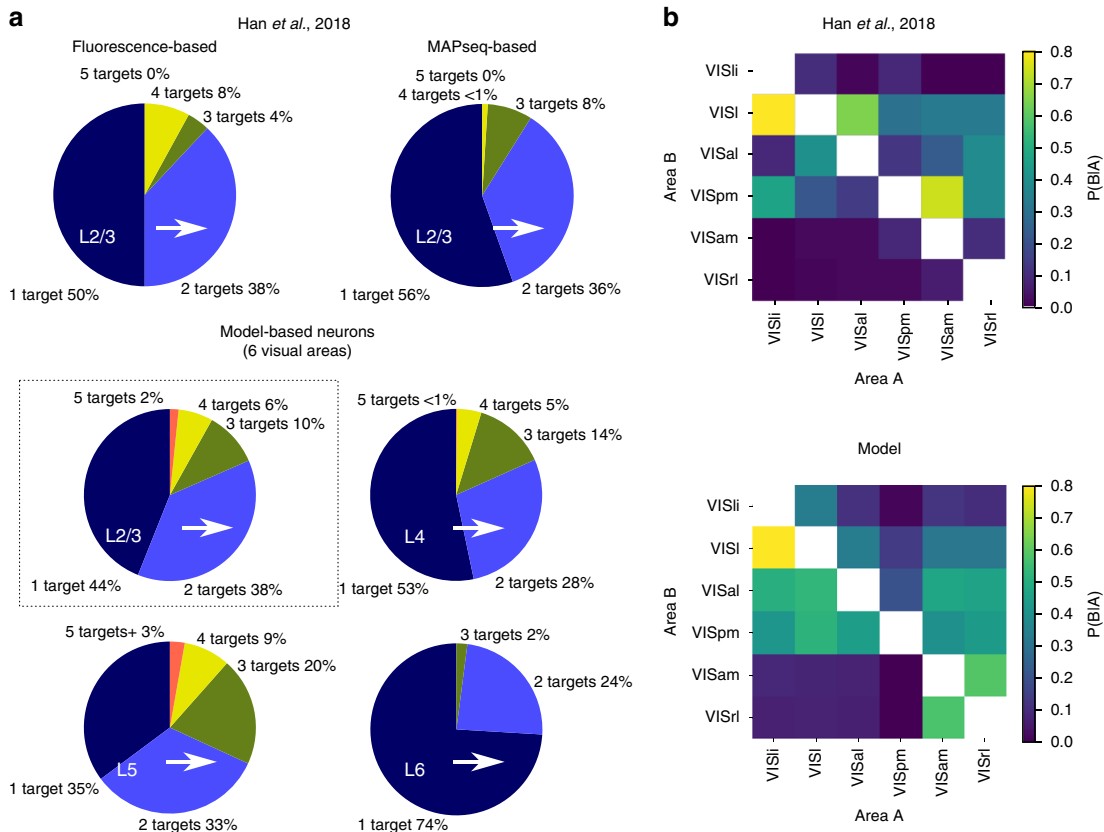

**Fig. 7** Validation of the tree-model. Validation against the results of Han et al.[31]. **a** Top, results of[31] in terms of the number of visual areas innervated by single axons originating in layer 2/3 of VISp. Bottom, corresponding results of the tree-model for axons originating in layers 2/3, 4, 5, and 6 (top left to bottom right; $n = 10,000$ innervation profiles each). **b** Top, results of Han et al.[31] in terms of common innervation of pairs of visual brain areas by axons originating in layer 2/3 of VISp. Bottom, corresponding results of the tree-model, based on $n = 10,000$ innervation profiles

postsynaptic neuron, and an exact location on the postsynaptic morphology (Supplementary Fig. 6). This allowed us to analyze the microstructure emerging from the constraints we added on top of the matrix of connection strengths. While the additional constraints on layer profiles and topographical mapping were arguably on the meso- rather than microscale, and the p-types governed the targeting of regions rather than individual neurons, they were together likely to affect measurements of the microstructure.

For example, an overexpression of reciprocally connected neuron pairs is traditionally a measure of microstructure[33,34]. Topographical mapping between regions *A* and *B* can lead to such an overexpression for pairs where one neuron is in *A* and the other in *B*. This occurs when a location in *A* is mapped to a location in *B* that is in turn mapped back to the same location in *A*, leading to reciprocal connectivity of neurons in those locations that is higher than expected from the average unidirectional probabilities between the regions. In order for this trend to emerge in an experiment, neurons would have to be sampled over sufficiently large volumes for the mapping to have a significant effect.

We evaluated the strength of this effect in an exemplary pair of connected regions, VISa and VISam (Fig. 8). We calculated unidirectional and reciprocal connection probabilities between parts of the regions, where we first defined a subvolume of VISam with increasing radius, then found the center of its projection to VISa according to the mapping and defined a subvolume with the same radius around the center (Fig. 8a, sampling radius). We found that the connection probabilities decreased with increased radius, as more and more parts of the regions are considered that are not mapped to each other (Fig. 8b). However, the expected reciprocal connection probability obtained from multiplying the unidirectional probabilities fell off faster than the measured one. Indeed for all radii over 150 μm, the reciprocal overexpression, i.e., the measured divided by the expected reciprocal probability, was in three connectivity instances larger than one, reaching values as high as 2.5 for radii over 500 μm (Fig. 8c).

In addition, we found that measuring connection probabilities not at the center of the projection of the subvolume, but offset from it (Fig. 8a, sampling offset) lead to an overexpression of reciprocally connected pairs. For a sampling radius of 150 μm, we shifted the center of the subvolume in VISa in a random direction by various amounts, finding that it decreased all connection probabilities while simultaneously leading to an increase in the reciprocal overexpression (Fig. 8d, e).

Motif counts in neuron triplets is another traditional measure of microstructure[33,34]; its equivalent in long-range connectivity is motif counts in triplets where each neuron is in a different brain region. The p-types dictate that certain pairs of regions tend to be innervated together, which would lead to overexpression of the corresponding motifs.

Using the same method of sampling from subvolumes as above (Supplementary Fig. 5a), we performed such an analysis for three regions that are strongly connected to each other, FRP, MOs, and MOp, confirming the trend. Based on 100,000 triplets in the subvolumes, we found that motifs where a neuron in FRP innervates only a neuron in MOp or a neuron in MOs only innervates a neuron in FRP where significantly underexpressed in favor of motifs where they innervate neurons in both other regions (Supplementary Fig. 5b).

The constraints on topographical mapping and the p-types are specific implementations of a principle of structured connectivity on various levels; not only between modules and regions, but also successively smaller subregions, leading to a scale-invariant structure, previously identified in human MRI data[20]. The topographical mapping generates a structure of subregions, as

outlined above, while the p-types generate larger structures of groups of regions that tend to be innervated together. As such, the the micro-connectome instances can be thought of as extending this principle—so far demonstrated for voxelized connectivity—further down to the level of individual neurons. Taylor et al.[20] quantified this structure for diffusion imaging voxels by detecting modules in the internal connectivity structure of two contiguous brain regions, and then considering the connectivity between the brain regions in terms of connection strengths between pairs of such within-area modules. They found that the distribution of strengths was much wider than in a random control, indicating that the within-area modules also structure the connectivity between areas. We replicated this experiment on the microstructure, i.e., the predicted neuron-to-neuron connection matrices within and between VISa and VISam (Fig. 8f, g). Upon grouping individual neurons in the two regions into 93 (VISa) and 179 (VISam) within-area modules and comparing the connectivity between them to a random control preserving individual neuron in- and out-degrees, we found comparable results. Repeating the analysis for all sufficiently strong projections (Fig. 8h), we found the same, predicting that the principle extends down to the level of individual neurons.

Taken together, we conclude that the constraints and principles we identified lead to a highly nonrandom microstructure of connectivity. While the structure is a prediction that will need to be validated, this demonstrates the utility of generating statistical connectome instances, as they reveal and quantify the interactions between mesoscale and micro-scale connectivity.

## Discussion

We have developed a way to generate statistical instances of a whole-neocortex mouse micro-connectome. This approach takes into account the current state of knowledge on region-to-region connectivity strengths, the laminar pattern of projection synapses, the structure of topographical mapping between regions and the logic of regional targeting of individual projection axons as derived from over 100 whole-brain axon reconstructions, and a comprehensive mesoscale model of projections, built from thousands of experiments[3]. Combining these data with a morphologically detailed model of neocortex[21] has allowed us to statistically predict connections with sub-cellular resolution, i.e., including the the locations of individual synapses on dendritic trees. Our approach is timely, as it leverages and integrates three very recent, publicly available data sets. Furthermore, its flexibility and modularity will allow it to readily use future data sets in place and in addition to the currently used ones. The resulting wiring diagram allows fundamental questions to be addressed, such as the nature and dynamics of clinically relevant brain rhythms as well as hierarchical interactions in the cortex, which are fundamental for understanding cortical coding and whole-brain regional dynamics.

As the available data on this topic remains sparse, our approach was as follows: we considered the formation of the connectivity as a stochastic process selecting one out of a space of possible wiring diagrams, and then sought out biological principles and rules that consecutively restrict this space of biologically viable wiring diagrams.

The principles we identified were not only based on the biological data but also a number of assumptions. The assumptions were necessary to break down the scale of the problem, to interpret the data (data assumptions) and structure it into principles (structuring assumptions), to formulate principles mathematically (modeling assumption), and to apply them to infer missing data (generalizing assumption). In order to interpret the resulting micro-connectome and predictions, one needs to first

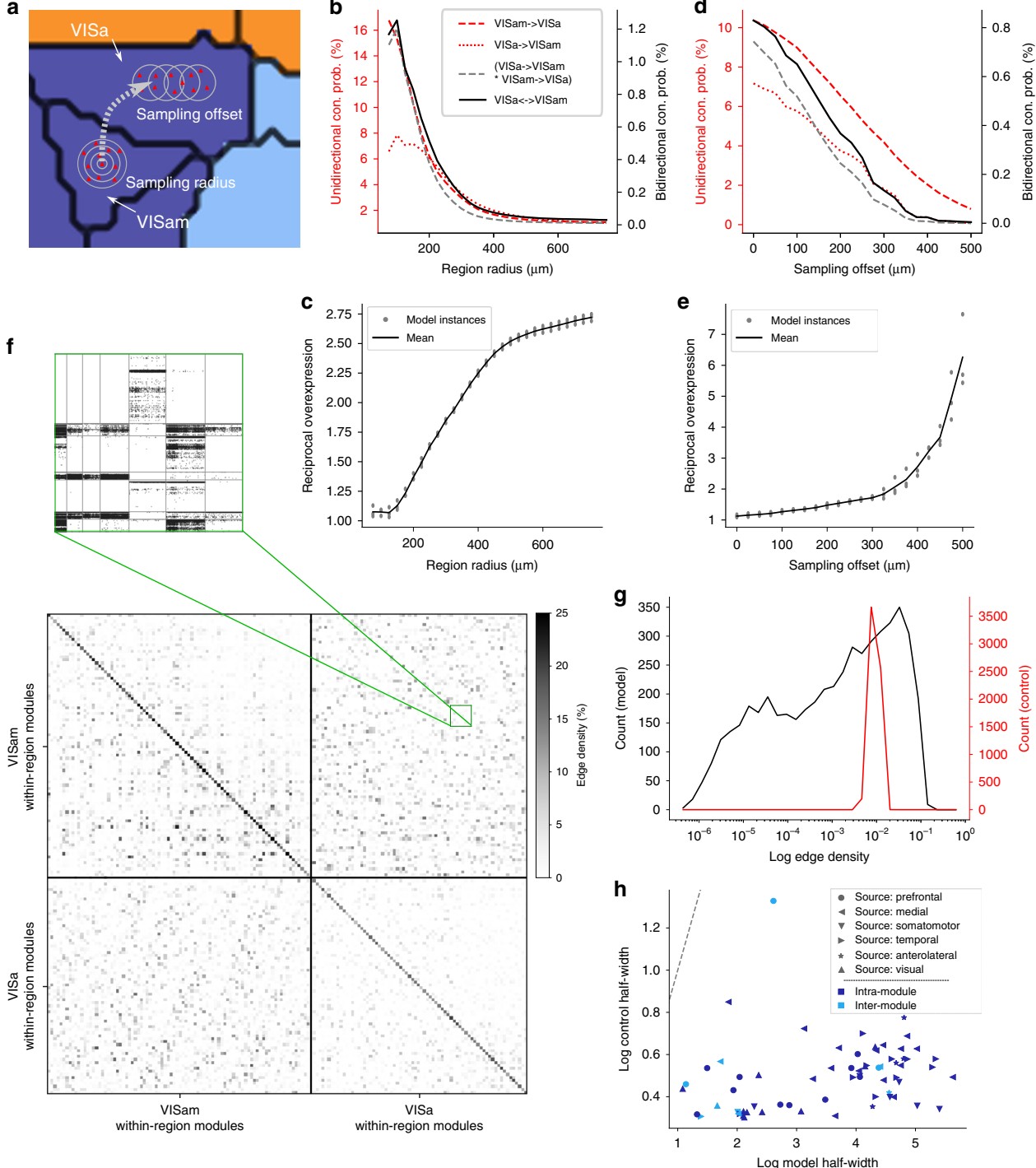

**Fig. 8** Bidirectional micro-connectivity and modularity. **a** Connectivity between individual neurons in VISa and VISam was sampled by defining a subvolume with various radii in VISam (sampling radius), then by finding the center of the projection from the subvolume to VISa according to the mapping (dashed arrow), moving it (sampling offset) and defining a subvolume with the same radius around it. **b** Unidirectional (red) and reciprocal connection probabilities for various sampling radii with zero-sampling offset. Gray: expected from unidirectional connectivity; black: model. **c** Ratio of reciprocal connectivity measured in the model over the expected value. Gray: three instances; black: mean of $n = 3$ instances. **d** As **b**, but for a sampling radius of 150 μm with various sampling offsets. **e** As **c**, but for sampling offsets. **f** Bottom: edge density, i.e., the number of connections over the number of pairs, of the microconnectivity between within-region modules that were defined by clustering the connectivity within the two brain regions (see the Methods section). Top: neuron-to-neuron connectivity between 7 × 7 within-region modules outlined in green. Gray lines indicate boundaries between within-region modules. **g** Distribution of edge densities in **f** (top right quadrant) compared with a random control. **h** Width of the distribution of edge densities (as in **g**) at half height, model against control, for projections with a density over 0.02 μm$^{-1}$. Circles: projection originating in the prefrontal module; stars: anterolateral module; left-pointing triangles: medial; downward-pointing triangles: somatomotor; right pointing: temporal; upward pointing: visual; dark blue: intramodule projection; light blue: intermodule

| Table 2 List of assumptions used in the formulation of the model | |
| --- | --- |
| Data assumption | Connectivity is symmetrical between hemispheres, both within and across hemispheres. |
| Data assumption | The amount of fluorescent signal is directly proportional to axon length in a region and the density of synapses per unit axon length is uniform across all neocortex. |
| Data assumption | Connection matrices for individual projection classes are version of the wild-type matrix, where submatrices are scaled by individual values, and that sum up to the wild type matrix. This assumption could be removed with more complete data on the projection classes. |
| Data assumption generalizing | First-order innervation probabilities for single axons can be predicted from the projection strength of the whole population in the source region. Validated in motor regions, we generalize the principle to all source regions. |
| Structuring assumption | The five projection classes considered are suitable to describe and parameterize long-range connectivity. |
| Structuring assumption | Each projection follows one of the six "prototype" layer profiles. |
| Structuring assumption | A clear distinction between local connectivity within a region and long-range connectivity across region, where local connectivity follows the principles outlined in Reimann et al.[24] and long-range connectivity the principles outlined here. |
| Structuring assumption | The brain parcellation scheme of the Allen Common Coordinate Framework is suitable to describe and parameterize the long-range connectivity. |
| Modeling assumption validated | The mapping of connections between regions is generalized topographical, continuous, and any scaling is linear. However, there is no assumption that it completely cover the source or target region. Note that we have quantified the amount of detail lost due to this. |
| Modeling assumption generalizing | The targeting of brain regions by individual axons can be explained by a tree-based model, generated from projection strength matrices. Validated for source regions VISp, MOs and MOp, we generalize to all other regions. |
| Inherited assumptions | Assumptions that went into the construction of the voxelized connectivity model. |
| Inherited assumptions | Assumptions that went into the construction of the morphologically detailed microcircuit model of Markram et al.[15]. |
| Implicit assumption | The constraints on long-range connectivity we considered are complete, that is no other biological principles restrict the space of viable wiring diagrams. This is almost certainly not true. |

understand these assumptions. While we have made them explicit in this paper, they are also summarized in Table 2 and discussed in the Supplementary Discussion.

As in any model, there is the implicit assumption of completeness, that our model captures all pertinent biological principles. We make no claim that this is true. This assumption is formally necessary for us to achieve the following modeling goal: given the assumptions, find the most general model that completely describes the data. In this context, we have drastically improved the strength of the null model of the microstructure of long-range connectivity. Previously, the most general model of the data was the null model implicit in long-range connection matrices—that of unstructured connectivity beyond the region-to-region level—or with at most some layer targeting rules. We have not only systematically integrated the data on this level but also added constraints that lead to a nonrandom microstructure with testable predictions.

Comparing potential experimental data against our improved model will lead to a better interpretation of the results. For example, we have demonstrated that an increased reciprocal microconnectivity between regions does not only necessarily imply a mechanism selectively stabilizing such motifs but can to some degree be explained by the mechanisms leading to topographical mapping. We have further demonstrated that in the presence of strong mapping, reciprocity must be evaluated relative to the trends present in the mapping to be correctly understood.

Findings violating the naive, unstructured null model but in line with our improved model can be explained by the principles of connectivity we implemented. For data points invalidating the model, for example conflicting triplet motif counts, we can try to pinpoint which assumption it violates and thus provide it context. Alternatively, data contradicting the model can be simply a result of biological variability between individuals. At this stage, we positioned the model to represent an average adult mouse where such false positives are least likely. Further, some constraints—such as the mapping and p-types—remained statistical and consequently captured a large degree of variability between individual instances. For the other constraints—such as average

synapse density and layer profiles—we can estimate an upper bound on variability in the future by running our programmatic pipeline to parameterize connectome constraints on outlier data points instead of averaged data. Similarly, other ages or specific strains can be modeled by using different data in the same pipeline.

We can already hypothesize about additional principles that might have to be added in the future. In terms of targeting of connectivity, we have implemented many aspects of spatial targeting of brain regions and locations within a region, and we have demonstrated that this leads to a highly nonrandom microstructure. However, it is possible that similar rules apply for the incoming long-range projections, i.e., which set of brain regions individual neurons are innervated by, and possible interactions between incoming and outgoing. In that case, we will be able to extent our definition of p-types to be the concatenation of incoming p-types and outgoing p-types.

In terms of the large-scale inter-area connectivity trends, i.e., the macro-connectome, our approach does not make any predictions, but is instead explicitly recreating the input data used. While Harris et al.[3] provided sufficient data for five projection classes, it missed for example a GABAergic projection class[18]. Additional sources could be used in the future to add such a type. In principle, completely different data sets could be used to define projection strengths. For example, Gămănuţ et al.[2] report a cortical mouse macro-connectome that recreates biological trends, such as a lognormal distribution over several orders of magnitude of projection strengths. They argue that their data captures several projections that are missed by Oh et al.[25] (and consequently also potentially by Harris et al.[3], which is based on similar computational methods). As their data provides potential sub-area resolution (see their Fig. S2), it could be used to also constrain the mapping and consequently serve as the basis of a stochastic micro-connectome predicted with our method, albeit without distinction of projection classes.

The assumption of a continuous, linear mapping between regions appears to solidly recreate the projection data, with only three regions leading to significant error (Fig. 4; MOs, MOp, and SSs). One explanation for the error would be that these regions

contain subregions that each send and receive their own, continuous projections. Indeed, for the projections from SSp-ll and SSp-ul to SSs (Supplementary Fig. 3b, right), we see several peaks of the green and blue color channels in the data, whereas a single continuous mapping can only generate single peaks. This is not surprising, as MOs, MOp, and SSs are not broken up by body part, unlike SSp that it strongly interacts with. In the future, the projection data could thus be used to further break up these regions, at least for the purpose of analyzing projections. With more advanced analyses and more data it may even become possible to hypothesize a brain parcellation scheme ab initio based on projection data.

Even with the imperfections outlined above, the present model will lead to advances in our understanding of brain function, when employed in simulations of whole-neocortex activity. The explicit parameterization of the constraints will allow us to change parameters to assess their impact. For example, it is at this point unclear whether the targeting rules for individual axons (p-types) will have an effect on high-level brain activity. Similarly, we can investigate to what degree the relatively simple topographical mapping in the model is sufficient for the upstream propagation of spatial information from VISp. Steps into that direction can be undertaken both in morphologically detailed models and point neuron models using the publicly available model connectome.

## Methods

### Accessing the mouse connectivity model
Unless noted otherwise, the data from the voxelized mouse connectivity model of the Allen Institute was accessed using the *mcmodels* python package provided by the authors (https://github.com/AllenInstitute/mouse_connectivity_models.git).

### Volumetric synapse densities of projections
We formulated a target mean density of synapses of $0.72\ \mu m^{-3}$ in the model, as measured by Schüz and Palm[26]. Multiplied with the neocortex volume of the Allen mouse brain atlas ($123.2\ mm^3$), this yielded a target number of 88.74 billion synapses. From this number, we subtracted 36 billion synapses we predicted in local connectivity within a brain region. This local connectivity was predicted by detecting axo-dendritic appositions in the model and filtering them to fulfill biological constraints, such as bouton density and synapses per connection[24]. We then derived a matrix of synapse densities in all projections between pairs of brain region by scaling the wild-type connection density matrices provided by Harris et al.[3] in the following way:

Let $M_i$ and $M_c$ be the $43 \times 43$ matrices of connection densities in ipsilateral and contralateral projections between brain regions, provided by Harris et al.[3]. Entries along the main diagonal of $M_i$, corresponding to connectivity within a region are set to 0. Furthermore, let $V$ be the vector of region volumes and $C_t$ the matrix of target region coverage in Supplementary Fig. 3d. Then we can calculate the scaling factor $\sigma$:

$$\sigma \cdot \sum_{a,b} (M_i[a,b] + M_c[a,b]) \cdot V[b] \cdot C_t[a,b] = 68.74 \cdot 10^9 \qquad (6)$$

This factor was then applied to both $M_i$ and $M_c$ to convert them into matrices of the average density of synapses in the target region due to a projection, measured in $\mu m^{-3}$. While this left no explicit room for synapses from extracortical sources, we estimate them to contribute comparatively little. For example, the density of thalamic synapses projected from VPM into SSp-bfd[27], when averaged over the whole-cortical depth, is only about 1.5% of the average total density ($0.72\ \mu m^{-3}$)[26].

### Projection density matrices for individual projection types
We combined the wild-type projection matrix from Harris et al.[3] with their incomplete information on projections in individual projection classes, to get five individual projection matrices, one for each projection class. As their wild-type experiments affected neurons in all layers and classes of the source region, we assumed that the sum of synapse densities over projection classes is equal to the density for the wild-type. Furthermore, based on qualitative observations, we assumed that the region-to-region connection matrices for each projection class are versions to the wild-type matrix, where individual module-to-module submatrices are scaled by individual values. The modules were six groups of contiguous brain regions (prefrontal, anterolateral, somatomotor, visual, medial, and temporal) identified in Harris et al.[3]. This assumption means that connectivity trends between modules will be preserved for all projection classes, but more fine grained trends for regions within a module will simply replicate the overall trends observed in the wild-type matrix for all classes.

Based on these assumptions, we derived matrices of synapse densities for individual projection classes with the following algorithm. First, we digitized the available information for individual projection classes from the Harris paper using the following mapping to cre-lines: 2/3: Cux2-IRES-Cre; 4: Scnn1a-Tg3-Cre; 5it: Tlx3-Cre_PL56; 5 pt: A93-Tg1-Cre; 6: Ntsr1-Cre_GN220. Then we condensed the information into five $6 \times 6$ matrices of average projection strengths between modules and normalized results such that the sum of the five matrices is 1 for each entry. Finally, we generated full-size 43 by 43 matrices for each projection type by scaling module-to-module specific submatrices of the wild-type matrix by the corresponding entry in the condensed and normalized matrix (Supplementary Fig. 7).

To reduce the computational demand of generating connectome instances, we determined a minimal projection strength and removed projections weaker than the cutoff. The cutoff was calculated as $0.0006\ \mu m^{-3}$, such that <5% of projection synapses would be lost.

### Projection density assumed symmetrical for both hemispheres
As the data in Harris et al.[3] are focused on the right hemisphere, we assumed connectivity to be symmetrical between hemispheres to be able to model both of them. This lead to 5 (projection classes) × 43 (source regions) × 86 (ipsilateral and contralateral target regions) potential projections parameterized in terms of their strength by the data. However, we considered the 5 × 43 ipsilateral projections within the same region to be local connectivity, which we instead derived with our established approach[24]. A number of regions also lack layer 4, rendering projections in that projection class void.

### Predicting layer profiles
To assign one out of six layer profiles to each projection, we digitized the data on profile frequencies of Harris et al.[3] and combined it according to the process illustrated in Supplementary Fig. 8: first, for a source module we counted the number of intra-module or inter-module projections originating from it in each projection class. The example illustrates inter-module feedforward projections from the prefrontal module (Supplementary Fig. 8, top left). For the presence of a projection, we defined a minimum projection strength, selected such that <5% of the total number of projection synapses are lost to the cutoff. The counts were then used as weights for a weighted average of the vectors of layer profile frequencies associated with each projection class. The result is a vector of expected profile frequencies for intra- or inter-module projections from the source module, if only the layer profile frequencies associated with projection classes are considered (Supplementary Fig. 8, top right).

Next, we looked up the observed profile frequencies for the source module in the data of ref. [3] and compared them to the expected ones (Supplementary Fig. 8, bottom left). Dividing the observed by the expected frequencies yielded adjustment factors for each layer profile that expressed which profiles were overexpressed or underexpressed in intra- or inter-module projections from the source module under consideration (Supplementary Fig. 8, bottom middle). We categorized projections as feedforward or feedback, based on the hierarchical positions of brain regions, reported in Fig. 8e of ref. [3], and, in accordance with their findings, reduced by 50% the adjustment factors for profiles 1, 3, and 5 when considering feedback projections and of profiles 2, 4, 6 when considering feed-forward projections. Finally, we multiplied the vector of adjustment factors with the vectors of profile frequencies for individual projection classes to get adjusted profile frequencies (Supplementary Fig. 8, bottom right).

The method yielded unique profile frequencies for each combination of source module, projection class and intra- or inter-module projection. To reduce the vectors of adjusted frequencies to a single profile, we simply picked the profile with the highest adjusted frequency (Supplementary Fig. 8, bottom right).

### Topographical mapping of projections
The topographical mapping of projections was defined by barycentric coordinate systems in the source and target regions and the assumption that a point in one region is mapped to the corresponding point in the other. The local coordinate systems were derived using the methods described in the Results section, implemented in custom python code available at: https://github.com/BlueBrain/Long-range-micro-connectome. However, due to the potentially large extent in the target region of single-projection axons, the biological mapping is rather point-to-area than point-to-point. Therefore, we additionally predicted for each projection the width of the targeted area.

A point-to-area mapping would result in an $N_{src}^{tgt}$ with lower saturation values, i.e., when depicted as in Fig. 3 in an image with slightly washed-out colors. Indeed, we found for most projections low saturation values in $N_{src}^{tgt}$ and consequently the optimal solution for the target coordinate system $M_{src}^{tgt}$ would place all three defining points outside the target region. However, we assumed that low saturation values were rather a result of a large extent of projection axons leading to a weak mapping. We therefore added another objective to the optimization procedure for $M_{src}^{tgt}$: minimizing the fraction of the source region that is mapped to points outside the target region. To compensate, we defined points in the source region to be mapped to 2d Gaussian kernels at their target location instead of a single point. The width of the Gaussian was optimized such that a convolution of $M_{src}^{tgt}$ with the same Gaussian resulted in the same distribution of saturation values as $N_{src}^{tgt}$.

**Analyzing whole-brain axons**. We acquired 183 neuron reconstructions from the Janelia Mouselight data portal[19] by querying for reconstructions where the soma location is within the neocortex. We first manually annotated the apical dendrite using Neurolucida (MBF Bioscience, Williston, VT, USA) given that it was not available in the original data. Based on this, we classified the neuron as a pyramidal cell or interneuron. Then, we performed a spatial analysis of the axon projection of each neuron by mapping the terminal points of the axon as well as the soma location into the Allen CCFv3 atlas coordinate system[25]. This yielded a complete list of brain regions containing axon terminal branches, as well as the brain region and layer containing the soma. Together with the information previously extracted from the annotated apical dendrite (e.g., shape, layer, number of branches), this spatial information is used to perform classification of the m-type and projection type (p-type) of the neuron.

**Testing statistical independence of region innervation**. Let $N$ be the number of analyzed axons (here: 61 for innervation from L5 of MOs). Let $n_a$ and $n_b$ be the number of them that innervate regions $a$ and $b$, respectively. Then under the assumption of statistical independence, the number of axons innervating both $a$ and $b$ is distributed according to the hypergeometric distribution with parameters $N$, $n_a$, $n_b$. We tested where the observed number of dual innervations fell along the cumulative distribution, and rejected the null hypothesis of independence if it was within the first or last 2.5% (two-tailed test).

**Constructing the p-type generating tree morphology**. The Louvain algorithm takes a weighted adjacency matrix as input, and then clusters the nodes into communities trying to maximize the weights within a community and minimize the weights across. An additional parameter is $\gamma$, which defines the granularity of the result: The smaller the value, the fewer communities it will result in, until a value of zero resulting in a single community.

We began by setting *gamma* to a value of 6.0, such that every brain region resulted in its own community. Correspondingly, we began constructing the tree topology by associating every brain region with its own leaf node. We then continuously lowered the value of $\gamma$, such that regions and communities began to merge into larger communities. We considered a pair of communities to be merged when through lowering $\gamma$ a new community appeared that contained more than half of the regions of each of the original communities. In that case, we placed a new node in the graph representing the new community and connected it with the two nodes representing the original communities. We continued lowering $\gamma$ until it reached zero, at which point everything merged into a single community and the root of the tree was placed.

We fit the weights of the edges to the predicted innervation probabilities using a recursive algorithm that optimized the local weights in small motifs consisting of two sibling nodes and their parent. It is based on the following observations (Supplementary Fig. 9):

Let $T_1$ and $T_2$ be two sibling nodes and $R$ their parent. In the model, any difference in the innervation probabilities for axons originating in $T_1$ and $T_2$ can only be due to differences in the lengths of the edges connecting each of them to their parent. This is because once the parent is reached, the shortest paths to any other region will be identical. Therefore:

$$w_{T_1 \to R} - w_{T_2 \to R} \approx |M[T_1, :]| - |M[T_2, :]| \qquad (7)$$

$$w_{R \to T_1} - w_{R \to T_2} \approx |M[:, T_1]| - |M[:, T_2]|, \qquad (8)$$

Where $M$ denotes the matrix of the negative logarithm of predicted innervation probabilities, $M[x, :]$ a single row of it (i.e., the probabilities of neurons in $x$ to innervate each other region), and $M[:, x]$ a single column of it (i.e., the probabilities of $x$ to be innervated by neurons in each other region).

Further, the probability a neuron in $T_1$ innervates $T_2$ is given by the path from $T_1$ via $R$ to $T_2$:

$$M[T_1, T_2] = w_{T_1 \to R} + w_{R \to T_2} \qquad (9)$$

$$M[T_2, T_1] = w_{T_2 \to R} + w_{R \to T_1} \qquad (10)$$

We found values for the four edge lengths in the motif by finding the least-squares solution of the system of linear equations. After this, we continued by performing the same step for node $R$ and its sibling, until the root was reached.

**Generating connectome instances according to the constraints**. As mentioned previously, a long-range projection recipe is created which describes constraints on the desired connectivity. By doing so, the same recipe can be used to instantiate long-range projections with different circuit models, and to allow for different implementations to create these instantiations. This section describes the implementation used to generate the connectomes published under https://portal.bluebrain.epfl.ch/resources/models/mouse-projections.

The circuit representation and input data required for this implementation are:

- A placement of neuron morphologies in space
- A table describe their morphological types
- A spatial index, allowing the querying of morphology segments in a bounding region
- An atlas describing the different regions and layers that are addressed by the recipe
- A "flat-mapping" from 3d to 2d space
- the recipe, which has:

  populations: defining which regions, subregions and morphological types are part of the various source and target populations
  projections: organized by source population; specifies per target population, the expected synapse density, layer profile, and the barycentric source and target triangles
  p-types: organized by source population; specifies per target population the first-order innervation probabilities for neurons of the source population, and for pairs of target populations the conditional increase in innervation probabilities

The basic circuit representation (first three items) was generated by a scaled-up version of a published algorithm[15]. The atlas was based on the Allen Common Coordinate Framework[35]. For the flat-mapping, we used the Allen Dorsal Flatmap of the mcmodels python package (see above). With this data, the implementation proceeds with the following steps:

*Neuron allocation*: For each source population, the neurons in those populations are allocated to participate in projections to a number of target populations according to specified fractions and statistical interactions. Where no interaction is specified, the overlap (neurons participating in both projections) is calculated from the fractions participating in one projection multiplied by the other; this default value is scaled-up, where interactions are specified. The challenge is then to assign neurons to each of the projections, such that the desired fractions and overlap sizes are reached.

A simplistic greedy algorithm was used to perform this allocation. Each source population group is assigned a sampled set of neurons, and pairwise the overlap is calculated, and adjusted based on the first-order interactions. When the overlap is too small, it is enforced by randomly sampling neurons from each group, and replacing neurons in the other group such that the overlap is achieved. Attempts were made to use a SAT solver to perform exact allocations, but the size of the neuron counts and the constraint counts meant the model could not be solved in the available memory.

*Synapse sampling*: Sampling happens at the target region level. The target populations in the region are grouped, and the required densities per incoming projection are computed based on the long-range projections recipe. The densities are translated into counts based on the constrained volume created by intersecting the area occupied by the barycentric triangles, and reversed mapped using the "flat-map" to the voxels of the atlas within the region. Finally, all morphological segments of a target population within this volume are found, and sampled with replacement with weights proportional to their length. Synapses are placed at random offsets within these segments. This structure allows for parallelization, as each combination of target and source populations can be run at the same time, subject to computation and memory limits.

In practice, finding all segments within the volume demands significant memory which constrains the implementation. This, in turn, gives rise to per target population order: all samples for this population are loaded, and then all the sources referencing this population are calculated sequentially, with the calculations parallelized when possible—the initial sampling, picking the segments with in the barycentric coordinates, etc. Further parallelization can be achieved by running many of these process on different machines, in a batch style.

*Mapping*: Following the allocation and sampling, the two results are brought together in mapping: source neurons that are allocated to a projection are matched to the synapses created during the sampling of the same projection. Because both these data sets work with 3d coordinates, they are projected into the 2d representation so that the barycentric coordinates, described earlier, can be used to create the desired spatial organization.

To that end, since source neurons are less numerous, they are first projected into the flat space, and from there mapped into the barycentric coordinate system of the source region. The same coordinates in the barycentric system of the target region are then mapped back into the flat space and considered the mapped locations of the source neurons in the target region. Synapses in the target region are directly mapped to the flat space. Finally, in parallel, synapses are then stochastically assigned to a target neuron with a weighting based on the distance to their mapped location in the flat space and the specified width of the mapping. To speed this process, the source locations are put in a k-dimensional tree, and only the 100 closest source locations are queried per potential target synapse.

*Output*: The final step is to output the circuit in a format that can be used for simulation. For this, the SONATA file format was chosen: https://github.com/AllenInstitute/sonata. In addition, for structural analysis, we output for each target region a connectivity matrix of all incoming connections in the scipy.sparse.csc_matrix format.

**Reporting summary**. Further information on research design is available in the Nature Research Reporting Summary linked to this article.

## Data availability

The recipe constraining the long-range connectivity—underlying Figs. 1–4, S1–S3—and stochastic instances fulfilling the constraints—underlying Figs. 6–8, S5 and S6—can be downloaded from the *Mouse whole-neocortex connectome model* portal (https://portal.bluebrain.epfl.ch/resources/models/mouse-projections/). The reconstructions of individual axons—underlying Fig. 5—are available at the MouseLight project at Janelia, mouselight.janelia.org.

## Code availability

The model was constructed using python 2.7 with custom code available at https://github.com/BlueBrain/Long-range-micro-connectome.

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

## Acknowledgements

We thank Lida Kanari for help with the validation of projection types, Joseph Knox for assistance with the projection from the Allen common coordinate framework into a 2d plane, and Dimitri Rodarie & Max Nolte for feedback on the paper. This study was supported by general funding to the Blue Brain Project from the Swiss government's ETH Board of the Swiss Federal Institutes of Technology and by support to BBP as one of its research centers by the École polytechnique fédérale de Lausanne.

## Author contributions

M.W.R. developed, implemented, and validated the techniques to constrain and para-meterize the long-range micro-connectome, and to express the constraints in a machine-readable format. E.M. contributed to the development of these techniques. M.W.R. generated the figures, and wrote the paper. M.G. developed and implemented the techniques to generate micro-connectome instances in a neocortex model based on the constraints. M.W.R. and M.G. validated the micro-connectome instances. Y.S. and H.L. analyzed the axonal targeting of individual neurons from the Janelia MouseLight data set. E.M. and H.M. provided guidance and contributed to writing the paper.

## Additional information

**Competing interests:** The authors declare no competing interests.

