## [Peer Review File · Nature Communications]

REVIEWERS' COMMENTS:

Reviewer #1 (Remarks to the Author):

The authors have done a great job revising their manuscript, further clarifying novel findings, and demonstrating the utility and promise of this first version model in linking micro-, meso-, and macro-scale connectivity. I again commend the authors for the clear and concise descriptions of assumptions and technical details in construction of the model.

The addition of the data presented in Figure 8 is a much more “compelling” example of the power of this model to predict (and thus to test) general organizing principles of brain wiring across scales. I acknowledge and appreciate very much the counter-point to my previous suggestion on adding simulations of cortical activity “...that the power and utility of the model lies not only in potential to fuel simulations, but also in furthering our understanding of the underlying anatomy itself.”

I think the authors have now demonstrated impact, novelty, and robust technical methods in constructing this model, and that others in the field will use it as intended as a resource from which predictions can be made and empirical data compared.

Although I am now enthusiastic about the publication of this manuscript in Nature Communications, I have a few additional questions and comments that I think need to be addressed as a prerequisite.

1) Information on the specific Cre line datasets used from Harris et al. (2018) for the L2/3, L4, L5 IT and L5 PT lines was added in the revised Methods (line 40-41), but I am now wondering why only a subset of lines were selected for building these matrices. Specifically, Harris et al. (2018) show 2 lines for L2/3 (Cux2 and Sepw1), 3 lines for L4 (Nr5a1, Scnn1a-Tg3, Rorb), 4 lines for L5 PT (A93, Chrna2, Efr3a, Sim1) and 2 lines for L6 CT (Syt6, Ntsr1). What was the rationale for selecting only some of the lines? This seems particularly problematic for the L5 PT and L6 CT data as long-range projections from several cortical areas were not mapped at all using the lines indicated in your Methods (i.e. Chrna2 and Syt6).

2) How generalizable is the ‘scale-invariant’ topology rule? The data shown (in Figure 8) are from only one probed connection (VISa and VISam) in one cortical module. Additional examples would further strengthen confidence in this prediction (or show whether this is unique to visual areas).

3) It still seems premature to apply the p-type rules to all cortical areas given the lack of data outside MOs L5 and VISp L2/3. Although this assumption is explicitly stated and described, I still have concerns about the validity of generalizing from the available single cell data. For example, when comparing between these two areas, there is a notable difference in the number of targets contacted by a single cell (in VISp L2/3: 50-56% neurons contact only one target, at least within visual areas; whereas in MOs, from Fig 5a, it appears that most neurons contact multiple targets, even if just considering the somatomotor module). This difference might be attributed to layer (as suggested by results in Figure 7), but it might also be a regional difference. How would this impact the predictions, e.g., of micro-connectome reciprocal topological wiring, if each node in a pair had a different set of p-type rules?

Reviewer #2 (Remarks to the Author):

As per my original review I believe the current journal is significantly more appropriate for the this manuscript.

In this revised version the authors have addressed my concerns. In particular the figures are now clearer, and the emphasis on the nonrandom connectivity clarifies the predictions/value of the paper.

Point-by-point reply to reviewers' requests

Reviewer 1:

The authors have done a great job revising their manuscript, further clarifying novel findings, and demonstrating the utility and promise of this first version model in linking micro-, meso-, and macro-scale connectivity. I again commend the authors for the clear and concise descriptions of assumptions and technical details in construction of the model.

The addition of the data presented in Figure 8 is a much more "compelling" example of the power of this model to predict (and thus to test) general organizing principles of brain wiring across scales. I acknowledge and appreciate very much the counter-point to my previous suggestion on adding simulations of cortical activity "...that the power and utility of the model lies not only in potential to fuel simulations, but also in furthering our understanding of the underlying anatomy itself."

I think the authors have now demonstrated impact, novelty, and robust technical methods in constructing this model, and that others in the field will use it as intended as a resource from which predictions can be made and empirical data compared.

Although I am now enthusiastic about the publication of this manuscript in Nature Communications, I have a few additional questions and comments that I think need to be addressed as a prerequisite. We thank the reviewer for their enthusiastic endorsement of our manuscript.

1) Information on the specific Cre line datasets used from Harris et al. (2018) for the L2/3, L4, L5 IT and L5 PT lines was added in the revised Methods (line 40-41), but I am now wondering why only a subset of lines were selected for building these matrices. Specifically, Harris et al. (2018) show 2 lines for L2/3 (Cux2 and Sepw1), 3 lines for L4 (Nr5a1, Scnn1a-Tg3, Rorb), 4 lines for L5 PT (A93, Chrna2, Efr3a, Sim1) and 2 lines for L6 CT (Syt6, Ntsr1). What was the rationale for selecting only some of the lines? This seems particularly problematic for the L5 PT and L6 CT data as long-range projections from several cortical areas were not mapped at all using the lines indicated in your Methods (i.e. Chrna2 and Syt6). We thank the reviewer for pointing out the inconsistency. Indeed, the lines we originally listed were from an earlier attempt, where we focused on only the prefrontal module. The actual list is: 2/3: Cux2-IRES-Cre; 4: Scnn1a-Tg3-Cre; 5it: Tlx3-Cre_PL56; 5pt: A93-Tg1-Cre; 6: Ntsr1-Cre_GN220. We have updated that in the manuscript accordingly. We simply picked for each class the line with the most experiments associated with them (as counted in Harris et al. Fig. 1a)

2) How generalizable is the 'scale-invariant' topology rule? The data shown (in Figure 8) are from only one probed connection (VISa and VISam) in one cortical module. Additional examples would further strengthen confidence in this prediction (or show whether this is unique to visual areas).

We thank the reviewer for pointing out the opportunity to extend our analysis to other regions and modules. We have done so and added the result as an additional panel (h) to Fig. 8 and point out the generalizability of the result in the manuscript:

"Repeating the analysis for all sufficiently strong projections (Fig. 8h), we found the same, predicting that the principle extends down to the level of individual neurons."

3) It still seems premature to apply the p-type rules to all cortical areas given the lack of data outside MOs L5 and VISp L2/3. Although this assumption is explicitly stated and described, I still have concerns about the validity of generalizing from the available single cell data. For example, when comparing between these two areas, there is a notable difference in the number of targets contacted by a single cell (in VISp L2/3: 50-56% neurons contact only one target, at least within visual areas; whereas in MOs, from Fig 5a, it appears that most neurons contact multiple targets, even if just considering the somatomotor module). This difference might be attributed to layer (as suggested by

results in Figure 7), but it might also be a regional difference. How would this impact the predictions, e.g., of micro-connectome reciprocal topological wiring, if each node in a pair had a different set of p-type rules?

The generalization power of the p-type rule, beyond the two validated source regions – indeed remains to be validated further in the future. As the reviewer points out, the model does indeed cover layer-specific differences, as it uses one tree structure per projection class. However, we would like to point out that it can also cover – and indeed predicts! – regional differences. The model allows for the lengths of edges in the tree to be different in different directions, i.e. A->B can be substantially shorter than B->A. As such, a neurons in region where nearby inward (towards the root) pointing edges have short lengths (leading to a high probability to cross them) will innervate many other regions. At the same time, as the same edges pointing outward can have high lengths, leading to low probability of being innervated. This effect leads to region-specific differences, albeit within the limits imposed by the tree topology used.

We clarified this in the manuscript adding:

“Next, we replaced each edge with two directed edges, one in each direction. [...] As the pair of edges between nodes can have different associated probabilities, the predicted statistical interactions are not symmetric (see Supplementary Figure 4) and there can be region-specific differences in the number of regions innervated or innervated from.”

Reviewer 2:

As per my original review I believe the current journal is significantly more appropriate for the this manuscript.

In this revised version the authors have addressed my concerns. In particular the figures are now clearer, and the emphasis on the nonrandom connectivity clarifies the predictions/value of the paper.

We thank the reviewer for this positive review of our manuscript.